# Prevalence and risk factors for antimicrobial resistance among newborns with gram-negative sepsis

Semaria Solomon[1,☯], Oluwasefunmi Akeju[2,☯], Oludare A. Odumade[3], Rozina Ambachew[1], Zenebe Gebreyohannes[1], Kimi Van Wickle[2], Mahlet Abayneh[1], Gesit Metaferia[1], Maria J. Carvalho[4,5], Kathryn Thomson[4], Kirsty Sands[4,6], Timothy R. Walsh[4,7], Rebecca Milton[4,8], Frederick G. B. Goddard[2], Delayehu Bekele[1,2], Grace J. Chan[1,2,3]*

1 St. Paul's Hospital Millennium Medical College, Addis Ababa, Ethiopia, 2 Harvard T.H. Chan School of Public Health, Boston, Massachusetts, United States of America, 3 Boston Children's Hospital, Harvard Medical School, Boston, Massachusetts, United States of America, 4 Division of Infection and Immunity, Cardiff University, Cardiff, United Kingdom, 5 Department of Medical Sciences, Institute of Biomedicine, University of Aveiro, Aveiro, Portugal, 6 Department of Zoology, University of Oxford, Oxford, United Kingdom, 7 Department of Zoology, Ineos Oxford Institute of Antimicrobial Research, University of Oxford, Oxford, United Kingdom, 8 Centre for Trials Research, Cardiff University, Cardiff, United Kingdom

☯ These authors contributed equally to this work.
* grace.chan@hsph.harvard.edu

**Data Availability Statement:** Sequencing data from our work have been deposited in the European Nucleotide Archive (ENA) under the

## Abstract

### Introduction

Newborn sepsis accounts for more than a third of neonatal deaths globally and one in five neonatal deaths in Ethiopia. The first-line treatment recommended by WHO is the combination of gentamicin with ampicillin or benzylpenicillin. Gram-negative bacteria (GNB) are increasingly resistant to previously effective antibiotics.

### Objectives

Our goal was to estimate the prevalence of antibiotic-resistant gram-negative bacteremia and identify risk factors for antibiotic resistance, among newborns with GNB sepsis.

### Methods

At a tertiary hospital in Ethiopia, we enrolled a cohort pregnant women and their newborns, between March and December 2017. Newborns who were followed up until 60 days of life for clinical signs of sepsis. Among the newborns with clinical signs of sepsis, blood samples were cultured; bacterial species were identified and tested for antibiotic susceptibility. We described the prevalence of antibiotic resistance, identified newborn, maternal, and environmental factors associated with multidrug resistance (MDR), and combined resistance to ampicillin and gentamicin (AmpGen), using multivariable regression.

project number PRJEB33565. All other relevant data are within the manuscript.

**Funding:** This study was funded by the Bill & Melinda Gates Foundation https://www.gatesfoundation.org/ OPP1119772 (TRW GJC). GJC and OAO received funding from the Boston Children's Global Health Program https://www.childrenshospital.org/globalhealth. The funders had no role in study design, data collection and analysis, decision to publish, or preparation of the manuscript.

**Competing interests:** The authors have declared that no competing interests exist.

## Results

Of the 119 newborns with gram-negative bacteremia, 80 (67%) were born preterm and 82 (70%) had early-onset sepsis. The most prevalent gram-negative species were *Klebsiella pneumoniae* 94 (79%) followed by *Escherichia coli* 10 (8%). Ampicillin resistance was found in 113 cases (95%), cefotaxime 104 (87%), gentamicin 101 (85%), AmpGen 101 (85%), piperacillin-tazobactam 47 (39%), amikacin 10 (8.4%), and Imipenem 1 (0.8%). Prevalence of MDR was 88% (n = 105). Low birthweight and late-onset sepsis (LOS) were associated with higher risks of AmpGen-resistant infections. All-cause mortality was higher among newborns treated with ineffective antibiotics.

## Conclusion

There was significant resistance to current first-line antibiotics and cephalosporins. Additional data are needed from primary care and community settings. Amikacin and piperacillin-tazobactam had lower rates of resistance; however, context-specific assessments of their potential adverse effects, their local availability, and cost-effectiveness would be necessary before selecting a new first-line regimen to help guide clinical decision-making.

## Introduction

More than 2.4 million neonatal deaths occur each year globally, the majority of which occur in low and middle-income countries (LMIC) [1]. About one-third of newborn deaths are caused by systemic infections, also referred to as neonatal sepsis [2, 3]. Ethiopia currently has a high neonatal mortality rate, which in 2018 was estimated at 28 per 1,000 live births and newborn sepsis accounts for about one in five neonatal deaths [4, 5]. Gram-negative bacteria (GNB) are estimated to be responsible for up to two-thirds of neonatal sepsis in Ethiopia [6–10]. Previous research suggests that the most predominant GNB isolates found among newborns with sepsis are *Klebsiella pneumoniae* and *Escherichia coli (E. coli)* [7, 9]. GNB are becoming increasingly resistant to previously effective antibiotics as new resistance genes can be readily transferred across GNB by mobile genetic elements [11]. An example is the plasmid-mediated inter-genus transfer of the resistance genes for extended spectrum beta-lactamases, which has been observed between *Escherichia coli* and *Klebsiella pneumoniae* likewise among other organisms in the *Enterobacteriaceae* family [12]. As a result, neonatal sepsis caused by GNB is twice as fatal as neonatal sepsis arising from gram-positive bacteria (GPB) [13–16]. The first line antibiotic therapy recommended by World Health Organization (WHO) to treat neonatal sepsis is a combination of intravenous or intramuscular gentamicin and benzylpenicillin or ampicillin, which is also the most common treatment of sepsis in infants under two months old in Ethiopia, [17, 18]. Third-generation cephalosporins, such as cefotaxime, ceftriaxone, and ceftazidime, are frequently used alternatives when resistance to first-line antibiotics is suspected [19].

Inappropriate antibiotic treatment for sepsis, due to antimicrobial resistance (AMR), has been linked to increased neonatal mortality and up to 30% of deaths from neonatal sepsis have been attributed to AMR [15, 20]. There have been reports of increasing rates of resistance to first-line and alternative therapies in some sub-Saharan African (sSA) countries [21–24]. A study from Ethiopia reported 91% of *Klebsiella spp.* and 67% of *E. coli* were resistant to ampicillin, and 82% and 56% resistant to gentamicin, respectively [7]. Furthermore, two studies

reported high rates of resistance to third-generation cephalosporins; however, both findings came from observing very few GNB isolates (n = 24, n = 14) [7, 9]. Studies have reported multi-drug resistance (MDR), defined as acquired resistance to at least one agent from three or more categories of antibiotics, to be greater than 70% for GNB [7, 9, 10, 25, 26]. However, there is a paucity of data from Ethiopia on the sensitivity of organisms to carbapenems and other infrequently used antibiotics.

There are known risk factors for antimicrobial resistance. Preterm birth, prolonged rupture of membranes, maternal infections, and prolonged hospitalization are some of the previously identified risk factors for neonatal sepsis [27–30]. Frequent antibiotic use, poor sanitation and hygiene, and poor compliance with infection control practices have been associated with an increased incidence of AMR [31–33]. Host factors could specifically predispose a newborn to infections by antibiotic-resistant GNB. Several studies have reported an increased risk of neonatal infections by antibiotic-resistant pathogens with intrapartum exposure to ampicillin [34–38]. Penicillins are typically the recommended intrapartum antibiotic given to women with group B Streptococci (GBS) colonization to reduce the risk of early neonatal sepsis [39]. Currently, there are limited data on the association between the incidence of antibiotic-resistant neonatal infections and other maternally administered antibiotics during labor. Understanding this relationship may guide the selection of intrapartum antibiotics.

The objective of this study was to estimate the prevalence of single and multidrug phenotypic resistance to 19 antibiotics, among GNB isolates from newborns with sepsis in Ethiopia. This data along with the subsequent follow-up of newborns from birth through 60 days after birth were used to identify neonatal, maternal, and environmental risk factors for AMR among newborns with gram-negative sepsis. This study provides evidence to inform future decisions and recommendations for treatment of newborn sepsis in Ethiopia and LMIC countries.

## Methods

### Study design and study population

As part of a multi-country study, Burden of Antibiotic Resistance in Newborns from Developing Societies (BARNARDS), we enrolled a cohort of mothers and their newborns at time of birth and followed them through 60 days of life between March 2017 and December 2017 at St. Paul's Hospital Millennium Medical College (SPHMMC), Addis Ababa, Ethiopia. We included newborns delivered at SPHMMC (inborn) and those born elsewhere and received care at SPHMMC (outborn). Among the newborns with clinical signs of sepsis, we obtained blood cultures to test for bacterial and fungal growth. In this secondary data analysis of AMR, we included a cohort of newborns with laboratory confirmed gram-negative sepsis. We excluded newborns without laboratory-confirmed sepsis, newborns whose blood cultures did not yield GNB isolates, and newborns who had no results for antibiotic susceptibility testing on their isolates (Fig 1).

### Ethical approval

Mothers enrolled in the BARNARDS-Ethiopia study provided informed written consent for themselves and their newborns. The study protocol was reviewed and approved by the ethical review committees of St. Paul's Hospital Millennium Medical College, Boston Children's Hospital, the Harvard T.H. Chan School of Public Health.

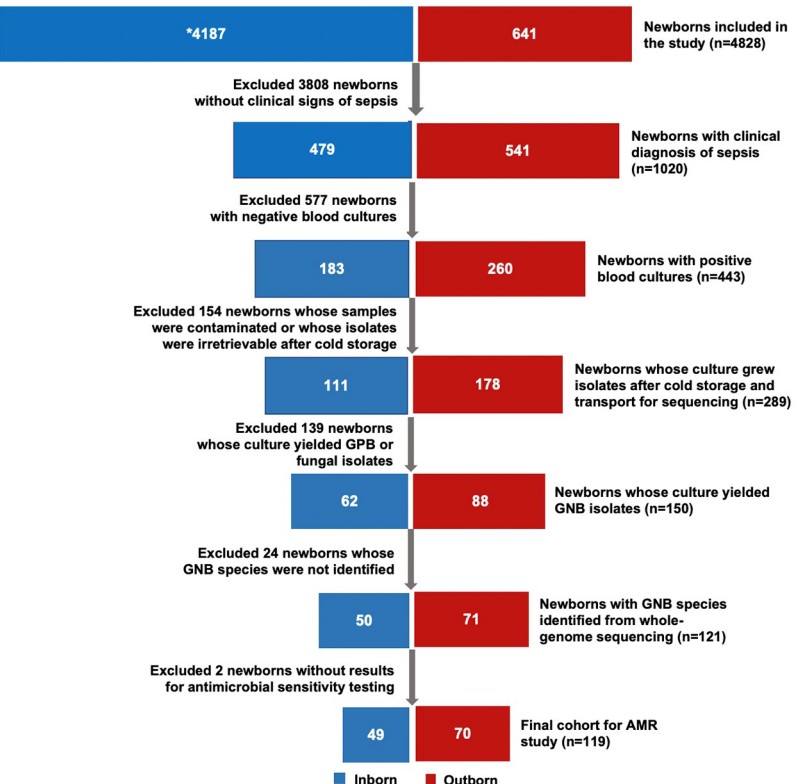

**Fig 1. Study cohort enrollment.** 4828 newborns were enrolled in the cohort. After excluding healthy newborns (no clinical signs of sepsis or negative blood cultures), newborns with blood cultures with contaminates, fungal or GPB isolates, unidentified GNB, 119 newborns were included in the study. GPB: gram-positive bacteria; GNB: gram-negative bacteria.

## Exposures and outcomes

The primary outcome of interest was the prevalence of resistant phenotypes of GNB isolates to 19 antibiotics, distinct antibiotic combinations, and multiple antibiotics. The nine classes of antibiotics assessed were penicillins (ampicillin, amoxicillin/clavulanic acid, piperacillin/tazobactam); cephalosporins (cefotaxime, ceftriaxone, ceftazidime, cefepime); carbapenems (meropenem, imipenem, and ertapenem); monobactams (aztreonam); aminoglycosides (gentamicin, amikacin, and tobramycin); fluoroquinolones (ciprofloxacin and levofloxacin); tetracyclines (tigecycline); fosfomycin; and polymyxin (colistin). An isolate was considered to be resistant to a class of antibiotics when it was resistant to at least one antimicrobial agent within that class [26]. Based on these nine classes of antibiotics, we created an MDR score that represented the number of antibiotic classes considered in our study that a GNB isolate is resistant to, on a scale of 0–9.

We considered ampicillin and gentamicin, ampicillin and cefotaxime, and piperacillin-tazobactam and amikacin as combinations, based on the high efficacy reported in a previous study [19, 40]. Resistance to an antibiotic combination was assumed when an isolate demonstrated resistance *in vitro* to each of the antibiotics that constitute the combination. We compared the 28-day and 60-day all-cause mortality rates between newborns who had been treated with antibiotic combinations to which the pathogen was later determined to be resistant and newborns who received antibiotics to which their infection was susceptible.

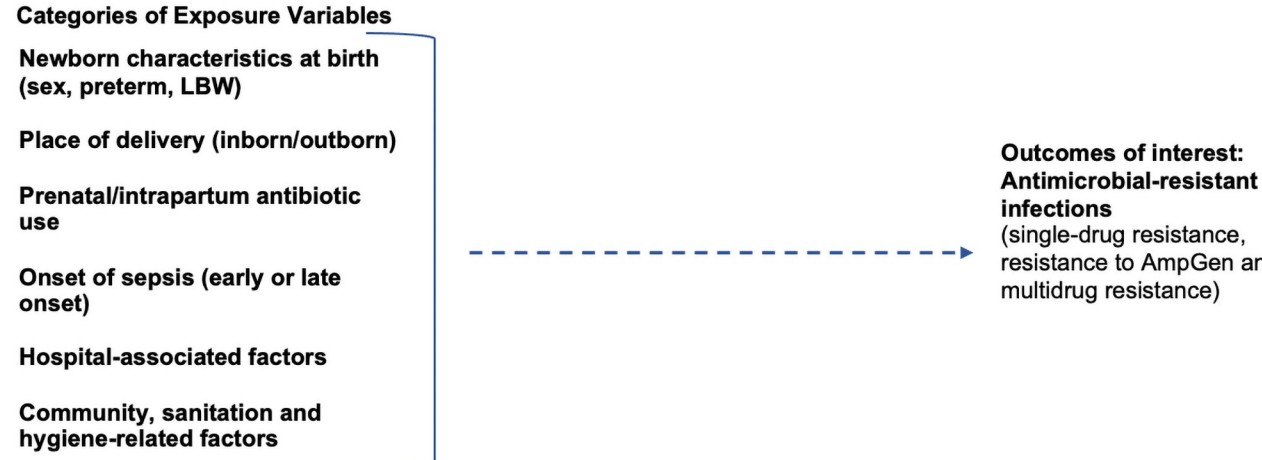

**Fig 2. Conceptual framework.** The conceptual framework describes the associations assessed in this study between exposures variables and the outcome of interest: antimicrobial-resistant infections.

To assess the association between exposure to intrapartum antibiotics and AMR in neonatal sepsis, we defined the exposure as any antibiotic administered to mothers during labor and delivery. The outcome was the prevalence of single-drug resistance to the intrapartum antibiotic among newborns with gram-negative sepsis.

In the analysis of the AMR risk factors, the exposures were categorized as:

1. Neonatal characteristics: birth cohort (inborn, outborn); sex (male, female); birthweight as binary (Normal $\geq$2500g; Low <2500g) and categorical (normal $\geq$2500g, low 1500g to <2500g, very low <1500g); gestational age as binary (term $\geq$37 weeks, preterm <37 weeks), and categorical (term $\geq$37 weeks, moderate to late preterm 32 to <37 weeks, very preterm <32 weeks); and type of delivery (vaginal or caesarean).

2. Onset of clinical sepsis (early-onset sepsis or EOS <72 hours of life, late-onset sepsis or LOS $\geq$72 hours to 28 days).

3. Factors relating to maternal antibiotic usage: prenatal antibiotic use during the last three months of pregnancy (yes, no); intrapartum antibiotic use (yes, no).

4. Sanitation and hygiene factors: type of toilet in the home (standard flush toilet, squat toilet/pit latrine, communal toilet outside the home/others); and access to running water (less than once/week, irregular or 2-3times/week, $\geq$4-6times/week).

The outcomes were resistance to both ampicillin and gentamicin (AmpGen) and MDR score (0–9). Fig 2 highlights the conceptual framework showing the associations between exposure and outcome variables.

## Data collection

Research staff collected standard of care data on maternal and neonatal clinical parameters from clinical records, interviews and direct observation of mothers and newborns. Study data collectors conducted home visits on the 3rd, 7th, and 28th days of life to examine newborns for symptoms of neonatal sepsis and phone visit on day 60 for maternal report of symptoms and

vital status outcomes. Neonatal blood samples for cultures were collected, stored, and analyzed according to best practices.

## Laboratory analysis

Upon clinical signs of sepsis, a neonatal blood sample was collected and subjected to analysis using an automated blood culture system (BACTEC™, BD). Following a positive blood culture indication, an aliquot of blood was transferred to Columbia Blood Agar supplemented with 5% sterile blood and incubated at 37˚C overnight. Preliminary identification at SPHMMC was performed using Gram-staining and Enterosystem 18R (for Enterobacterales). All isolates were stored on charcoal swabs and kept at 2–8˚C for shipment to the Cardiff University, UK, as part of BARNARDS, and in compliance with UN3373 regulations. Thereafter, species identification and genomic characterization were performed using whole-genome sequencing and agar dilution was carried out to determine minimum inhibitory concentrations (MIC), the lowest concentration of an antibiotic that inhibits bacteria's growth. Phenotypes of isolates were classified according to the breakpoints for MIC, as recommended by the European Committee on Antimicrobial Susceptibility Testing (EUCAST) [41], as part of BARNARDS [42]. An isolate's phenotype was categorized as "S-Susceptible using standard dosing regimen" when there was a high likelihood of therapeutic success using a standard dosing regimen of the agent; "I-Susceptible with increased exposure" when there was a high likelihood of therapeutic success because exposure to the agent is increased by adjusting the dosing regimen or by its concentration at the site of infection; "R-Resistant" when there is a high likelihood of therapeutic failure even when there is increased exposure [41].

## Statistical analysis

We used descriptive statistics to summarize exposure variables. We estimated the prevalence of GNB isolates resistant to each of 19 antibiotics tested and selected antibiotic combinations and the prevalence of MDR. Using a one-sided test of proportion, we assessed whether treating neonatal sepsis with antibiotic combinations that the causative GNB were resistant to, was associated with increased all-cause mortality at 28 and 60 days of life. We used a one-sided test of proportion to examine whether the prevalence of resistant phenotypes to each intrapartum antibiotic used in this study was higher among newborns with a history of maternal exposure to the specific antibiotics.

Since gram-negative bacteria are associated with an increased risk of AMR (Fig 2), we conducted our study within the subgroup of newborns with GNB sepsis when identifying risk factors for AMR sepsis. We used multivariable regression models to identify and adjust for possible sources of potential confounding among exposure variables.

We assessed the relationship between exposure variables and AmpGen using bivariable and multivariable log-binomial regressions. We assessed the association between exposure variables and MDR score using bivariable and multivariable linear regressions. We selected covariates for the multivariable models using the purposeful selection method [43]. We included candidate covariates with p-values <0.25 from the bivariable analysis in a multivariable model and those with a p-value of >0.1 were dropped if their exclusion did not result in >20% change in the effect estimate of other covariates. We added covariates with p-value >0.25 in the bivariable analysis if they had p-values <0.1 in the multivariable model. We excluded covariates with ≥20% observations missing from all analyses. We analyzed all data using STATA 16.1 and set statistical significance at a p-value of <0.05.

## Results

### Summary characteristics

Between March 2017 and December 2017, as part of BARNARDS-Ethiopia, 4,589 mothers and their 4,828 newborns (inborn = 4,583, outborn = 705) were enrolled and samples were obtained for blood cultures from 1,020 newborns (inborn = 479; outborn = 541) with clinical signs of sepsis. There were 443 newborns (inborn = 183; outborn = 260) with positive blood cultures. Of these, 300 isolates retrieved from samples of 289 newborns revealed the prevalence of GNB sepsis as 50% (n = 150), GPB 47% (n = 141) and fungal infections 3% (n = 9). Of the 11 newborns with more than one pathogen, four had a GNB and an unspecified GPB; one had both GPB and fungus while the additional pair of isolates or species in the remaining were duplicates of the original. Whole-genome sequencing was used to identify 121 GNB species from isolates belonging to 121 newborns. Two newborns were excluded because of insufficient data on AST (Fig 1). Our cohort for the AMR study consisted of 119 newborns, 70 (59%) outborn, and 49 (41%) inborn.

Of the 119 newborns, 53 (45%) were females, 53 (45%) were males, and sex was unreported for 13 (11%) newborns (Table 1). The majority of newborns (n = 80; 67%) were born preterm, of which 77 (96%) of the preterm newborns had low birthweight. Of the term newborns, 15 out of 39 (38%) were low birthweight. Early-onset sepsis (EOS) was identified in 56 out of 80 (70%) preterm and 26 out of 39 (67%) term newborns. For all newborns, the median time to clinical sepsis from delivery was one day (IQR = 0, 4 days). Nearly all (n = 118; 99%) newborns with GNB sepsis were hospitalized and blood samples had been collected from most of them (n = 99; 84%) for suspected sepsis prior to or on the day of hospitalization. The most predominant GNB species identified were *Klebsiella pneumoniae* (*Kp*) (n = 94; 79%), followed by *Escherichia coli* (n = 10; 8%), and *Acinetobacter baumannii* (n = 6; 5%) (Fig 3). There were 11 sequence types (ST) and capsular-antigen serotypes (KL) of *Kp* identified, the most predominant being ST35/KL108 (n = 38), ST37/KL15 (n = 28), ST218/KL57 (n = 10) and ST985/KL39 (n = 8), which were responsible for 89% (n = 84/94) of all *Kp* infections (Fig 4). Incidence of ST35 during the study period was similar for inborn and outborn while the incidence of ST37 was twice among inborn (33%) compared to outborn (17%) and eight of the ten newborns who had ST218 were outborn. Newborns delivered vaginally had higher incidence of ST35 and ST218 while increased incidence of ST37 was observed with Caesarean delivery (Fig 5). New infections by other *Kp* strains and non-*Kp* GNB were either sporadic (when rare) or fairly distributed over the entire study period, while ST35, ST37, ST218, and ST985 were more clustered, with at least 70% of cases occurring within two months (Fig 6).

### Prevalence of AMR

Among the penicillin class of antibiotics, ampicillin-resistance was observed in 113 (95%) isolates while resistance to piperacillin-tazobactam was found in 47 (39%) isolates (Fig 7). For aminoglycosides, resistance was high to gentamicin (n = 101; 85%) but very low to amikacin (n = 1; <1%). There was a high prevalence of resistance to all third and fourth-generation cephalosporins tested. Resistance to ceftriaxone was found in 105 (88%) isolates, cefotaxime 104 (87%), ceftazidime 104 (87%), and cefepime 101 (85%). Resistance to ciprofloxacin, a fluoroquinolone, was observed in 50 (42%) isolates. Nearly all isolates were susceptible to imipenem (n = 118; 99%) and meropenem (n = 118; 99%). All isolates were susceptible to colistin. Most isolates were resistant to AmpGen (n = 101; 85%), the recommended first-line therapy, likewise ampicillin and cefotaxime (n = 104; 87%), another antibiotic combination used for treating sepsis among study participants. Among newborns with

**Table 1. Prevalence and risk factors for antimicrobial resistance among 119 newborns with gram-negative sepsis.**

| Characteristics | N = 119 | Percentages (%) |
|---|---|---|
| **Median age at diagnosis in days for all newborns (IQR)** | 1 (0, 4) | |
| Median age at diagnosis for newborns with EOS (IQR) | 0 (0, 1) | |
| Median age at diagnosis for newborns with LOS (IQR) | 6 (4, 6) | |
| **Sex** | | |
| Female | 53 | 45 |
| Male | 53 | 45 |
| Missing | 13 | 11 |
| **Newborn Term** | | |
| Term (37-42weeks) | 39 | 33 |
| Preterm (<37weeks) | 80 | 67 |
| Moderate/Late Preterm (32-<37 weeks) | 62 | 52 |
| Very/Extremely Preterm (<32weeks) | 18 | 15 |
| **Birthweight** | | |
| Normal birthweight ($\geq$2500g) | 27 | 23 |
| Low birthweight (1500-<2500g) | 52 | 44 |
| Very low/Extremely low birthweight (<1500g) | 40 | 34 |
| **Birth Cohort** | | |
| Inborn | 49 | 41 |
| Outborn | 70 | 59 |
| **Type of Delivery** | | |
| Vaginal | 60 | 50 |
| Caesarian Section | 34 | 29 |
| Missing | 25 | 21 |
| **Type of Sepsis** | | |
| Early Onset | 82 | 69 |
| Late Onset | 35 | 29 |
| Missing | 2 | 2 |
| **Antibiotics Administered to Newborns** | | |
| Ampicillin & Gentamicin | 83 | 70 |
| Ampicillin & Cefotaxime | 11 | 9 |
| Ampicillin & Ceftriaxone | 1 | 1 |
| Vancomycin & Ceftazidime | 5 | 4 |
| No antibiotic received | 12 | 10 |
| Missing | 7 | 6 |
| **Intrapartum Antibiotics** | | |
| No | 72 | 61 |
| Yes | 25 | 21 |
| Missing | 22 | 18 |
| **Prenatal Antibiotics** | | |
| No | 101 | 85 |
| Yes | 4 | 3 |
| Missing | 14 | 12 |

EOS, AmpGen-resistance was higher among preterm (91%) than term (65%) newborns, and this finding was similar for MDR (Fig 8). Prevalence of MDR was 88% (n = 105) and the median number of antibiotic classes in which an isolate demonstrated resistance to at least one drug, was 4 (IQR = 4, 5).

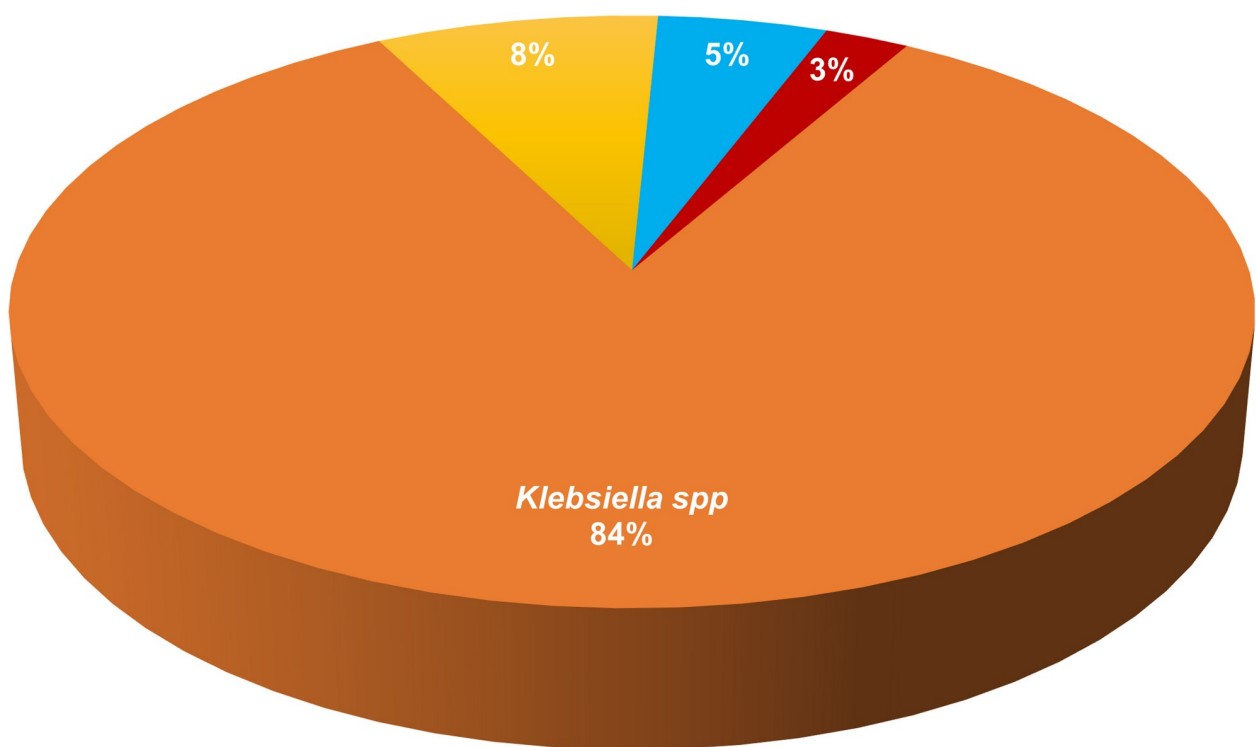

■ *Klebsiella spp (n=100)*

■ *Escherichia coli (n=10)*

■ *Acinetobacter baumannii (n=6)*

■ *Aeromonas spp (n=1); Citrobacter spp (n=1); Pseudomonas stutzeri (n=1)*

**Fig 3. Prevalence of gram-negative bacterial species.** Among the 119 newborns, the majority of gram-negative isolates identified were *Klebsiella spp*, followed by *Escherichia coli* and *Acinetobacter baumanni*.

## Antimicrobial therapy and treatment outcomes

Of the 119 newborns with blood culture confirmed sepsis, 100 (84%) received antibiotics. Antibiotic data were missing for seven newborns and 12 did not receive antibiotics due to death occurring shortly after admission, parental refusal of antibiotics, and discharge from hospital against medical advice. Ampicillin was administered to 95 out of 100 newborns who received antibiotics (Fig 9). Among the 83 newborns who were administered ampicillin and gentamicin, 72 (87%) had AmpGen-resistant infections. Eight out of the 11 newborns treated with ampicillin and cefotaxime had GNB phenotypes resistant to this antibiotic combination. Of 119 newborns with GNB sepsis, 30 (25%) died by their 28th day of life. At day 60, 72 (61%) were alive, 43 (36%) dead, and 4 (3%) were lost to follow-up (Figs 10 and 11). Compared to AmpGen-susceptible infections, AmpGen-resistant infections were associated with higher 28-day (n = 2/18, 11%; versus n = 28/97, 29%; p = 0.058) and 60-day (n = 4/18, 22%; versus n = 39/97, 40%; p = 0.074) all-cause mortality, although the differences observed were not statistically significant. The prevalence of 28-day and 60-day all-cause mortality was significantly higher among newborns who had received antibiotics to which their GNB infections were

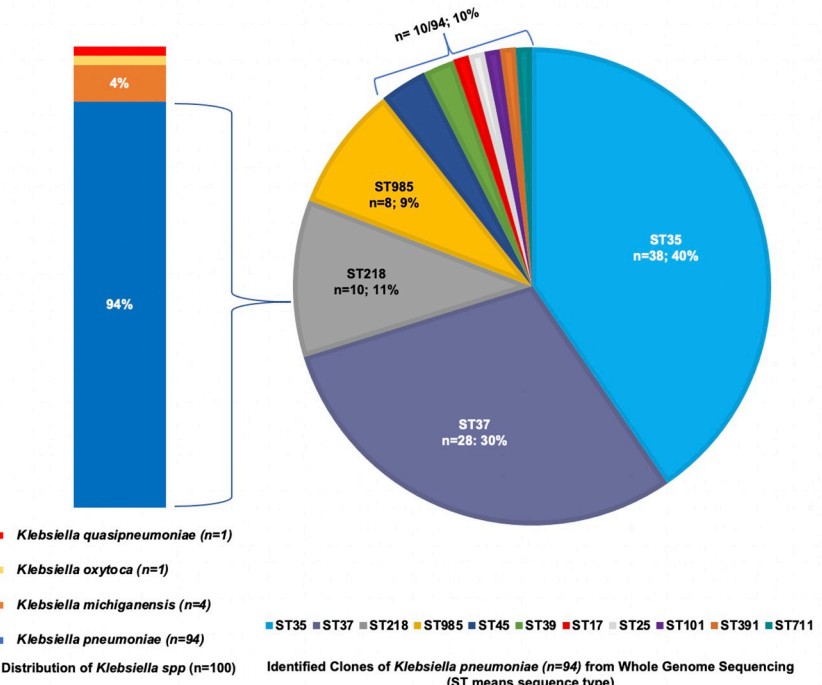

**Fig 4. Distribution of *Klebsiella spp* (n = 100) and the clonal strains of *Klebsiella pneumoniae* (n = 94).** *Klebsiella pneumoniae* accounted for 94 out of the 100 *Klebsiella spp* identified. There were 11 sequence types (ST) and capsular-antigen serotypes (KL) of *Klebsiella pneumoniae* identified; ST35/KL108 (n = 38) and ST37/KL15 (n = 28) accounted for 70% (n = 66/94) of all infections attributed to *Klebsiella pneumoniae*.

resistant (29% and 41%, respectively) compared to those whose treatment matched their infections (0% and 14%, respectively) (p = 0.010, p = 0.028). These findings were consistent among preterm and term newborns, inborn and outborn, newborns with EOS or LOS, regardless of the antibiotic combinations used. 28-day all-cause mortality among newborns who were treated with AmpGen for AmpGen-resistant infections (29%) was similar to that observed among newborns without evidence of receiving antibiotics (31%, p = 0.839). Nearly all the *Kp* strains (97%) were resistant to AmpGen, but the 60-day all-cause mortality among newborns infected with *Kp* strains were not much different from the average in this study except for ST218 where eight out of the 10 infected newborns died (Fig 12).

### Risk factors for AMR among all newborns

Of the 97 newborns with complete data, 25 (26%) were exposed to intrapartum antibiotics (Ampicillin, n = 11; and Ceftriaxone, n = 14). Intrapartum exposure to ampicillin (n = 11) and ceftriaxone (n = 14) were associated with a higher prevalence of infections with ampicillin-resistant (100% vs. 94%, p = 0.20) and ceftriaxone-resistant (93% vs. 86%, p = 0.24) GNB phenotypes, respectively (Fig 13); however, neither of these associations was statistically significant. The prevalence of prenatal antibiotic use was low (n = 4/119; 3%) in this study hence, its possible relationship with AMR was not assessed.

Compared to term newborns, preterm newborns had a 27% greater risk of infections with GNB that were resistant to AmpGen, (RR = 1.27; 95% CI = 1.03, 1.56) (Table 2). The risk of AmpGen resistance was 1.15 times (95% CI = 0.96, 1.39) higher among newborns from households whose access to running water was less than one day per week compared to newborns whose households had running water at least four days per week. The outborn did not have a

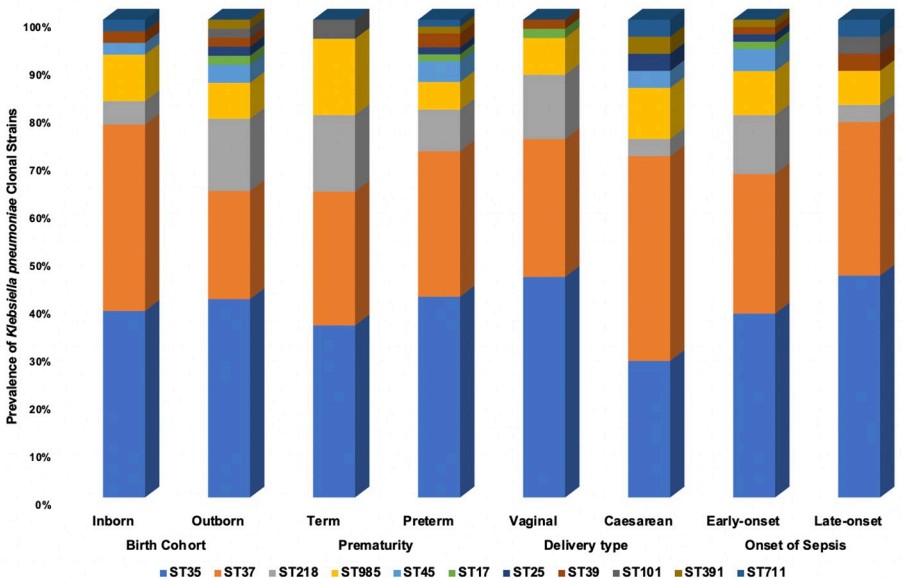

| Strains/Serotypes of Kp | | Birth Cohort | | Prematurity | | Delivery type | | Onset of Sepsis | |
|---|---|---|---|---|---|---|---|---|---|
| Sequence Types | Capsular (K) Antigen Serotypes | Inborn (n=41) | Outborn (n=53) | Term (n=25) | Preterm (n=69) | Vaginal (n=52) | Caesarean (n-28) | Early-onset (n=65) | Late-onset (n=28) |
| ST35 | KL108 | 39% | 42% | 36% | 42% | 46% | 29% | 38% | 46% |
| ST37 | KL15 | 39% | 23% | 28% | 30% | 29% | 43% | 29% | 32% |
| ST218 | KL57 | 5% | 15% | 16% | 9% | 13% | 4% | 12% | 4% |
| ST985 | KL39 | 10% | 8% | 16% | 6% | 8% | 11% | 9% | 7% |
| ST45 | KL24 | 2% | 4% | 0% | 4% | 0% | 4% | 5% | 0% |
| ST17 | KL112 | 0% | 2% | 0% | 1% | 2% | 0% | 2% | 0% |
| ST25 | KL2 | 0% | 2% | 0% | 1% | 0% | 4% | 2% | 0% |
| ST39 | KL62 | 2% | 2% | 0% | 3% | 2% | 0% | 2% | 4% |
| ST101 | KL106 | 0% | 2% | 4% | 0% | 0% | 0% | 0% | 4% |
| ST391 | KL30 | 0% | 2% | 0% | 1% | 0% | 4% | 2% | 0% |
| ST711 | KL54 | 2% | 0% | 0% | 1% | 0% | 4% | 0% | 4% |

**Fig 5. Distribution of the clonal strains of *Klebsiella pneumoniae* by location of birth (inborn or outborn).** The incidence of ST35 was similar for inborn and outborn newborns, however the incidence of ST37 was twice as high among inborn (33%) compared to outborn (17%) newborns. The incidence of ST218 was three times as high among outborn (15%) compared to inborn (5%) newborns.

significantly lower risk of AmpGen resistance compared to the inborn (RR = 0.94; 95% CI = 0.81, 1.10). The multivariable model for AmpGen resistance consisted of sex, LBW, and LOS. Preterm birth was excluded as it was found to be collinear with birthweight. After adjusting for low birthweight and LOS, the risk of AmpGen-resistant infections was 15% (RR = 0.85; 95% CI = 0.76, 0.96) lower among female newborns as compared to male newborns. Low birthweight and LOS were associated with 36% (RR = 1.36; 95% CI = 1.02, 1.83) and 13% (RR = 1.13; 95% CI = 1.03, 1.23) higher risks of AmpGen-resistant infections, respectively.

Newborns with LBW were more likely to develop GNB sepsis that are resistant to nearly one additional class of antibiotics, compared to those with normal birthweight (increase in MDR score = 0.95 points; 95% CI = 0.31, 1.59) (Table 3). LOS and birthweight were the only variables included in the multivariable analysis. With birthweight held constant, the MDR score for newborns who developed LOS was higher by 0.59 points (95% CI = 0.02,1.16) compared to newborns with EOS.

## Discussion

The majority of GNB infections among newborns enrolled in Addis Ababa, Ethiopia were resistant to the first-line antibiotics, ampicillin, and gentamicin, consistent with previous

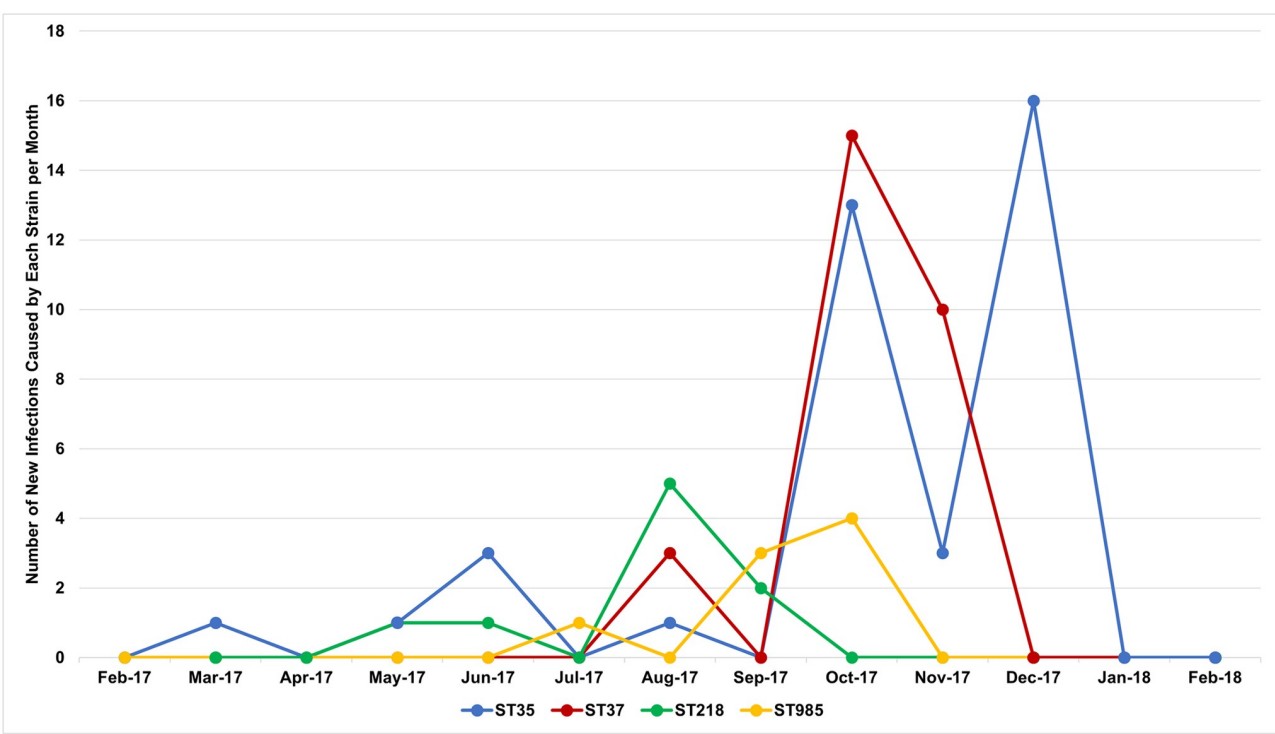

**Fig 6. Seasonal timing of *Klebsiella pneumoniae* strains.** Infections with *Klebsiella pneumoniae* strains, ST35, ST37, ST218, and ST985, were reported every month. Most of ST218 cases occurred between August and September; ST35 and ST37 cases occurred between October and December.

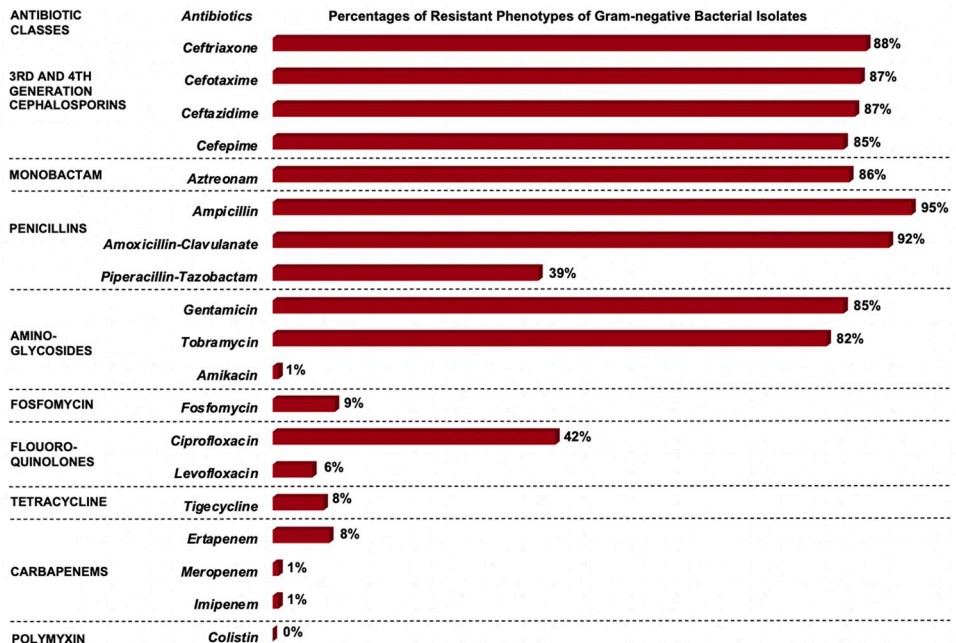

**Fig 7. Prevalence of gram-negative resistant phenotypes.** Over 80% of the 119 gram negative isolates were resistant to 3rd and 4th generation cephalosporins, monobactams, and penicillins (with the exception of piperacillin-tazobactim), and aminoglycosides (with the exception of amikacin). Less than 10% of isolates were resistant to amikacin, fosfomycin, levofloxacin, tigecycline, colistin, and all the carbapenems.

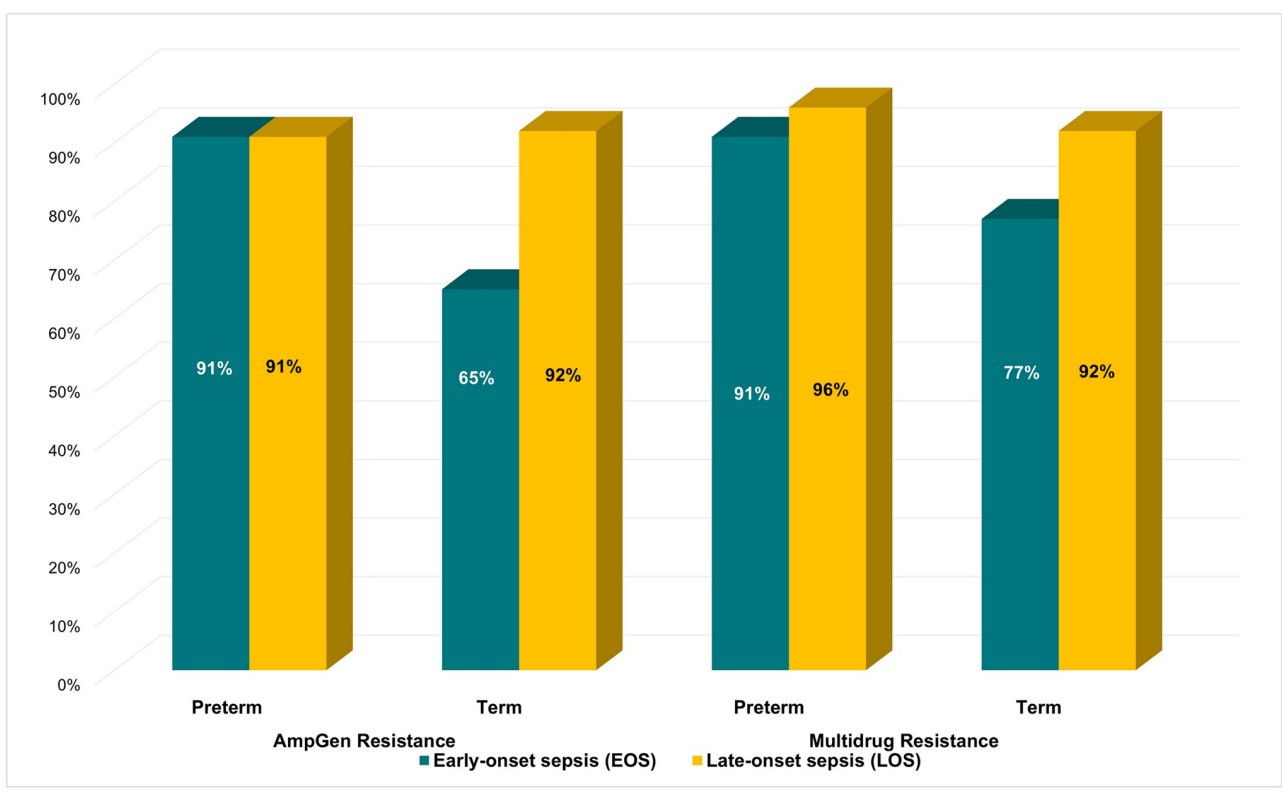

**Fig 8. Prevalence of MDR and AmpGen resistance by timing of sepsis (early versus late onset-sepsis) and gestational age (preterm versus term).**
Among newborns with early-onset sepsis, AmpGen resistance was higher in those born preterm (91%) compared to term newborns (65%). AmpGen resistance and MDR were higher among term newborns who had late-onset sepsis. MDR: multidrug resistance; GNB: gram-negative sepsis.

results of combined BARNARDS countries [42] (Sands *et al.*, in press; Thomson *et al.*, in press). This study assessed the extent of AMR against a wide range of antibiotics and identified potential alternatives with sensitive antimicrobial activity against GNB sepsis among newborns. In addition to seeking to improve the health outcomes of newborns diagnosed with GNB sepsis, the high prevalence of MDR in this study demonstrates the importance of addressing the growing threat of AMR in developing countries.

The high rates (>80%) of MDR to ampicillin, gentamicin, and the third-generation cephalosporins, the routine antibiotics for sepsis at health facilities, are concerning and consistent with recent findings in Ethiopia and across sub-Saharan Africa (sSA) [7, 24]. Resistance to amikacin and carbapenems was low (<1%) in this study. This is in line with findings from a meta-analysis which estimated the resistance of *Klebsiella spp*, the most predominant GNB, to amikacin and carbapenems to be 5% and 0% in East Africa, with 14% and 4% as the average for sSA, respectively [24]; however, the prevalence of resistance to amikacin and carbapenems were much higher in other regions. The resistance of GNB isolates to amikacin in North Africa and the Middle East was up to 88% while their prevalence of carbapenem-resistant GNB was as high as 94% [32, 44, 45]. These suggest regional differences in the distribution patterns of AMR. To the best of our knowledge, there was no prior study published from Ethiopia on the prevalence of piperacillin-tazobactam resistance among GNB isolates, which was 39% in this study. A pooled analysis from sSA reported a similar prevalence of piperacillin-tazobactam resistance among *Klebsiella spp.*, at 37% [24]. The high rate of resistance to routinely used drugs, which were once effective could serve as an indication that resistance to antibiotics

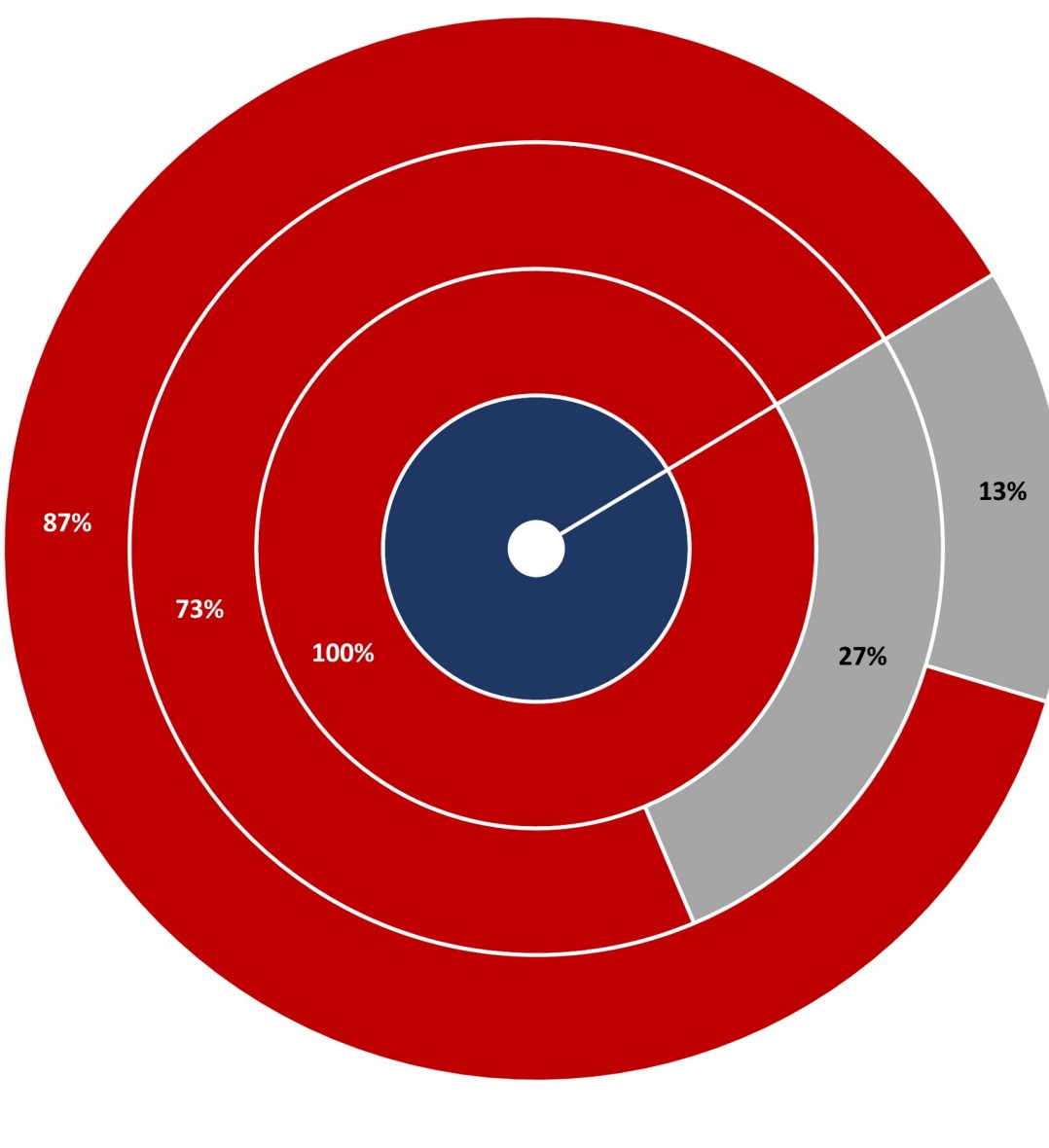

■ **Rightly matched antibiotics**   ■ **Wrongly matched antibiotics**   ■ **No antibiotic**

**Fig 9. Proportion of newborns with gram-negative infections resistant to the antibiotics received.** 87% (n = 72/83) of newborns who received ampicillin and gentamicin were resistant to this combination. 73% (n = 8/11) of newborns who received ampicillin and cefotaxime were resistant to this combination.

currently found to be efficacious against GNB may rapidly increase within a population if they become first-line therapy [21, 33]. This is further supported by the moderate rate of resistance to ciprofloxacin, an antibiotic, not usually used in children but frequently prescribed for treating infections among older individuals in Ethiopia [46]. This highlights the urgency for more judicious use of antibiotics across all age groups within a population.

The WHO recommends empirical treatment with antibiotics, based on clinical signs when neonatal sepsis is suspected, before laboratory confirmation of diagnosis [17] because of the rapid rate of disease progression and the limited access to laboratory evaluations in sSA countries that have the highest burden of neonatal sepsis [19]. The ideal empirical antibiotic would

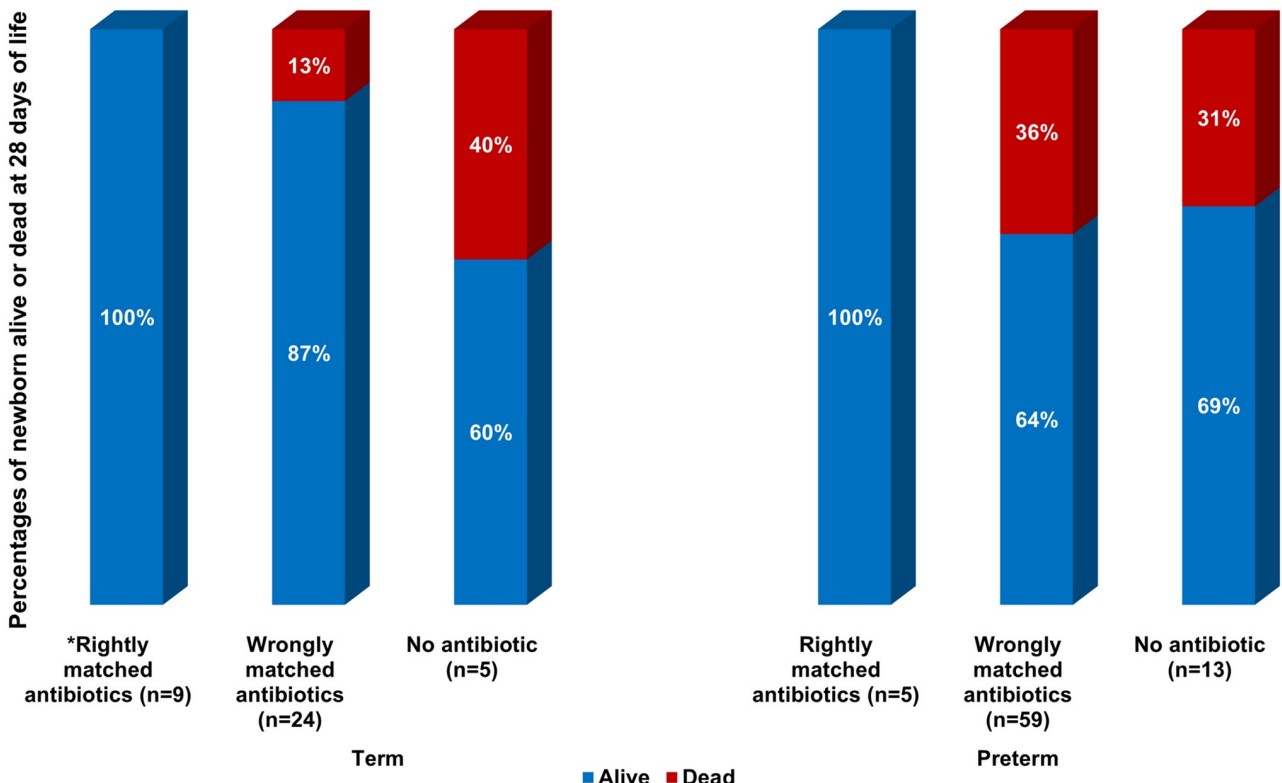

**Fig 10. Newborn mortality at 28 days among term and preterm newborns by antibiotic choice (rightly matched, wrongly matched, no antibiotics).** There were no deaths at 28 days among newborns who received antibiotics that were effective against their infections. Among newborns who received antibiotics to which their infections were resistant, the mortality rate was 36% (n = 21/58) among preterm newborns and 13% (3/24) among term newborns. *Rightly matched antibiotics represent antibiotic combinations for which the phenotypes of identified gram-negative isolates were susceptible. Wrongly matched antibiotics represent antibiotic combinations for which the phenotypes of identified gram-negative isolates were resistant. No antibiotic refers to no evidence or documentation that any antibiotics were received.

be inexpensive and would cover common GPB and GNB without increasing AMR [19]. As current evidence suggests the need for new treatment guidelines on antibiotics for neonatal sepsis, it may be necessary to consider regional differences in antibiotic sensitivity before choosing new empirical antibiotics; however, globalization may aid the transmission of antibiotic-resistant strains of GNB to areas where an antibiotic had not been introduced [47, 48]. Based on its high efficacy against GNB in this study and others, amikacin could be a potential drug to consider in Ethiopia and across sSA. Amikacin has also proven to be much more effective than gentamicin when used for treating neonatal sepsis in other countries, although they both share similar limitations of administration via intravenous route and the potential to harm the kidneys [19, 49]. Another alternative is piperacillin-tazobactam, which has a fair coverage of GNB, in Ethiopia and sSA, and has been associated with better treatment outcomes in practice than ampicillin or gentamicin [24, 50]. Despite demonstrating a good coverage of GNB sepsis in this study, the increasing global resistance and cost of carbapenems would prevent them from being recommended for widespread use; however, they may be feasible options for use as second-line therapy [51]. Tigecycline, fosfomycin, and levofloxacin also demonstrated high efficacy against GNB in this study and have been used as salvage therapies for extensively drug-resistant infections in newborns; however, levofloxacin [52] is not recommended for routine use because of its potential adverse effect on the musculoskeletal system of children while data on the safety, tolerability, and dosing of tigecycline [53] and fosfomycin

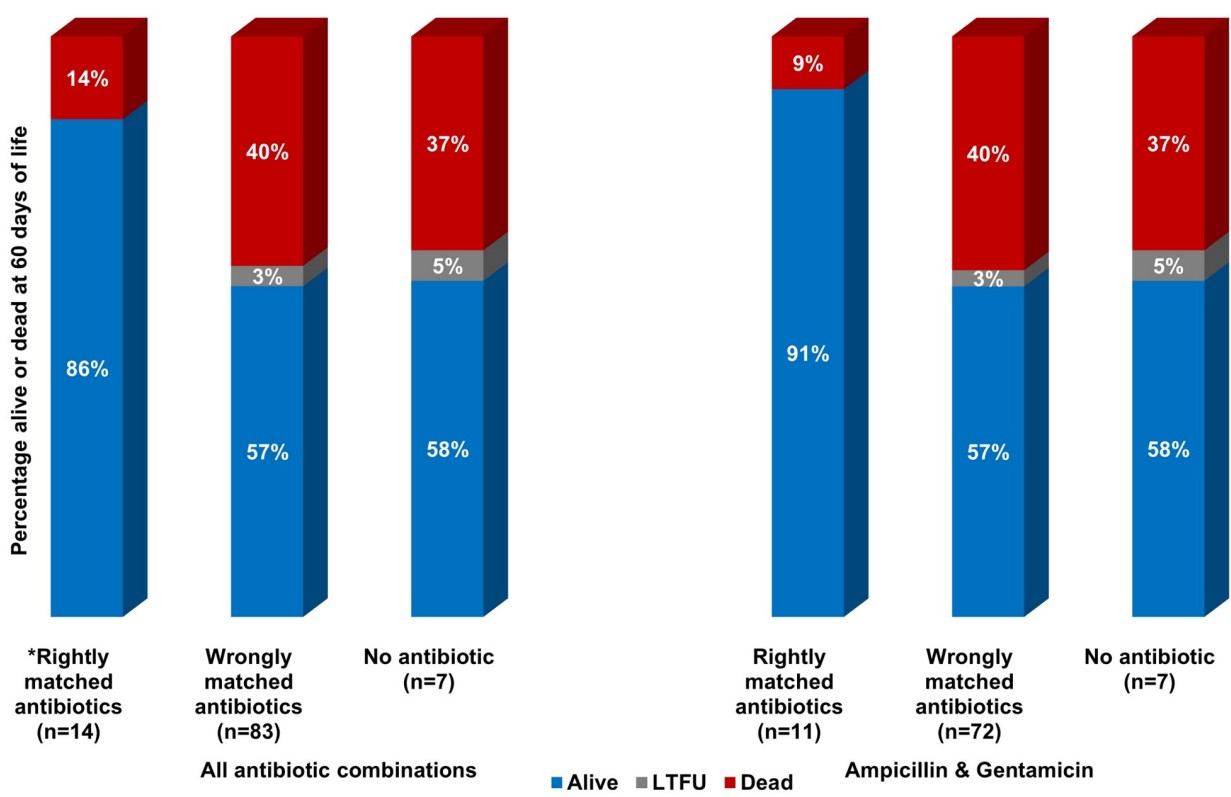

**Fig 11. Newborn mortality at 60 days among term and preterm newborns by antibiotic choice (rightly matched, wrongly matched, no antibiotics).** Among newborns who received antibiotics that were effective against their infections, there were two deaths (14%) out of 14 term newborns at 60 days, and 1 death (9%) out of 11 preterm newborns. Mortality at 60 days among term and preterm newborns who received antibiotics to which their infections were resistant was 40% and 37% among those who received no antibiotics. *Rightly matched antibiotics represent antibiotic combinations for which the phenotypes of identified gram-negative isolates were susceptible. Wrongly matched antibiotics represent antibiotic combinations for which the phenotypes of identified gram-negative isolates were resistant. No antibiotic refers to no evidence or documentation that any antibiotics were received. LTFU, lost to follow-up and vital status is unknown.

[54] in newborns are limited. Additional research is needed on the safety, pharmacokinetics, and cost-effectiveness of prospective sensitive antibiotics among newborns, especially those born preterm [55, 56]. In settings without microbiology capacity, treating newborns with clinical symptoms of sepsis with "big gun" antibiotics may do more harm than good by increasing antibiotic resistance or adverse side effects. Furthermore, the mortality outcome of these sensitive antibiotics should be taken from the experience of other countries utilizing amikacin or piperacillin-tazobactam as a first-line regimen.

Preterm birth and low birthweight (LBW) are well-known risk factors for neonatal sepsis. Hence, the likely explanation for the majority of newborns in our study being LBW, and most of the newborns with LBW in our study were born preterm. Preterm birth and LBW were also found to be significant risk factors for AmpGen resistance and MDR, especially among newborns with early-onset sepsis (EOS). Evidence supporting the increased risk of antibiotic-resistant infections with preterm birth was found in two studies [16, 32]. The less developed immune system among preterm newborns, generally thought to be one of the underlying reasons for increased sepsis among preterm newborns, does not directly explain the higher risk for antibiotic-resistant infections [29]. The majority of preterm newborns in this study had EOS which may be associated with maternal intrapartum transmission of pathogens that colonize the mothers' gastrointestinal or reproductive tract [28, 30]. Further research would be

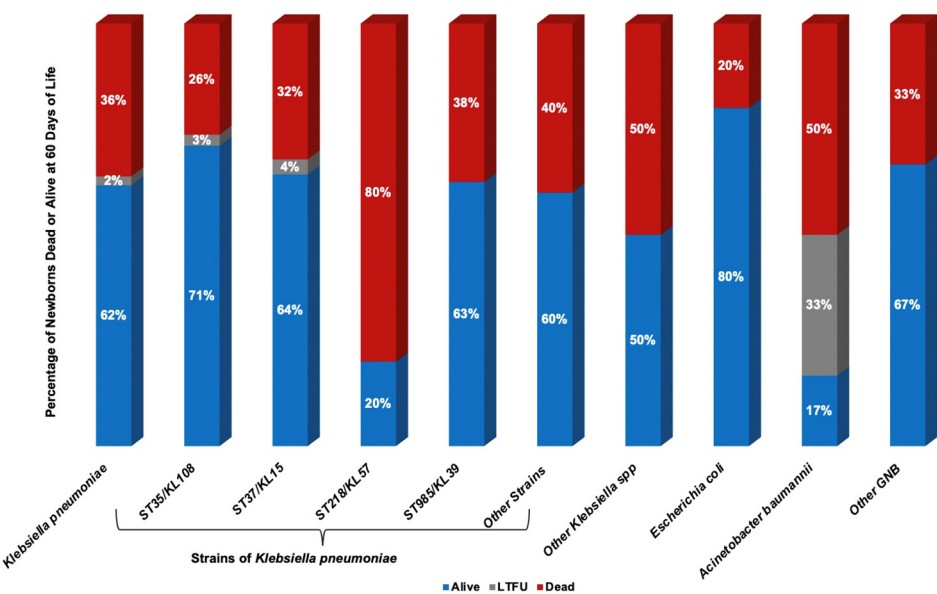

**Fig 12. Newborn mortality at 60 days by gram-negative species and strain.** The ST218 *Klebsiella pneumoniae* strain was the most virulent strain; 80% of newborns infected with ST218 died. GNB: gram-negative bacteria; MDR: multidrug resistance; MDR Score: number of antibiotic classes where an organism/strain is resistant to at least one antibiotic agent; LTFU: lost to follow-up.

| GNB Species or Sequence Types & K-Serotypes | Other Characteristics | | | | | Outcome at 60 Days of Life | | |
| --- | --- | --- | --- | --- | --- | --- | --- | --- |
| | Preterm | AmpGen Resistance | MDR | *MDR Score (IQR) | Wrong/ No Antibiotics | Alive | LTFU | Dead |
| *Klebsiella pneumoniae* (n=94) | 73% (69) | 97% (91) | 99% (93) | 4 (4, 5) | 98% (92) | 62% (58) | 2% (2) | 36% (34) |
| ST35/K108 (n=38) | 76% (29) | 97% (37) | 100% (38) | 4 (4, 4) | 100% (38) | 71% (20) | 3% (1) | 26% (10) |
| ST37/K15 (n=28) | 75% (21) | 93% (26) | 100% (28) | 5 (5, 5) | 93% (26) | 64% (18) | 4% (1) | 32% (9) |
| ST218/K57 (n=10) | 60% (6) | 100% (10) | 100% (10) | 4.5 (4, 5) | 100% (10) | 20% (2) | 0% (0) | 80% (8) |
| ST985/K39 (n=8) | 50% (4) | 100% (8) | 100% (8) | 5 (5, 5) | 100% (8) | 63% (5) | 0% (0) | 38% (3) |
| Other Strains (n=10) | 90% (9) | 100% (10) | 90% (9) | 4 (4, 5) | 100% (10) | 60% (4) | 0% (0) | 40% (4) |
| Other *Klebsiella spp* (n=6) | 50% (3) | 50% (3) | 67% (4) | 4.5 (1, 6) | 83% (5) | 50% (3) | 0% (0) | 50% (3) |
| *Escherichia coli* (n=10) | 40% (4) | 10% (1) | 20% (2) | 1 (0, 2) | 10% (1) | 80% (8) | 0% (0) | 20% (2) |
| *Acinetobacter baumannii* (n=6) | 67% (4) | 100% (6) | 100% (6) | 7.5 (7, 8) | 100% (6) | 17% (1) | 33% (2) | 50% (3) |
| Other GNB (n=3) | 0% (0) | 0% (0) | 33% (1) | 1 (1, 3) | 33% (1) | 67% (2) | 0% (0) | 1 (33%) |

needed to understand whether this increased risk of AMR is due to unexplained host factors among preterm newborns that select for resistant pathogens, or whether the mothers of preterm newborns and newborns with LBW are more likely to have a higher prevalence of colonization by antibiotic-resistant strains of GNB.

Late-onset sepsis (LOS) was associated with an increased risk of MDR in this study and the risk was not particularly higher among babies with LBW. This suggests that antibiotic-resistant GNB are likely to be more prevalent within the primary sources of LOS in Ethiopia. Unlike EOS, sources of infection in LOS have been attributed to be more from horizontal contamination post-delivery than vertical transmission from the mother during labor [27]. GNB infections in LOS can be community-acquired or hospital-acquired, especially when newborns are admitted in neonatal intensive care units [27]. Prevalence of antibiotic resistance is high among hospital-acquired GNB infections and with poor infection control practices [57]. Unsanitary conditions in the environment have also been reported as risk factors for AMR [58]. In this study, the risk of MDR was increased with unsanitary conditions such as lack of access to running water within households but the association did not reach statistical significance, possibly due to the limited sample size. Future research to understand the distribution

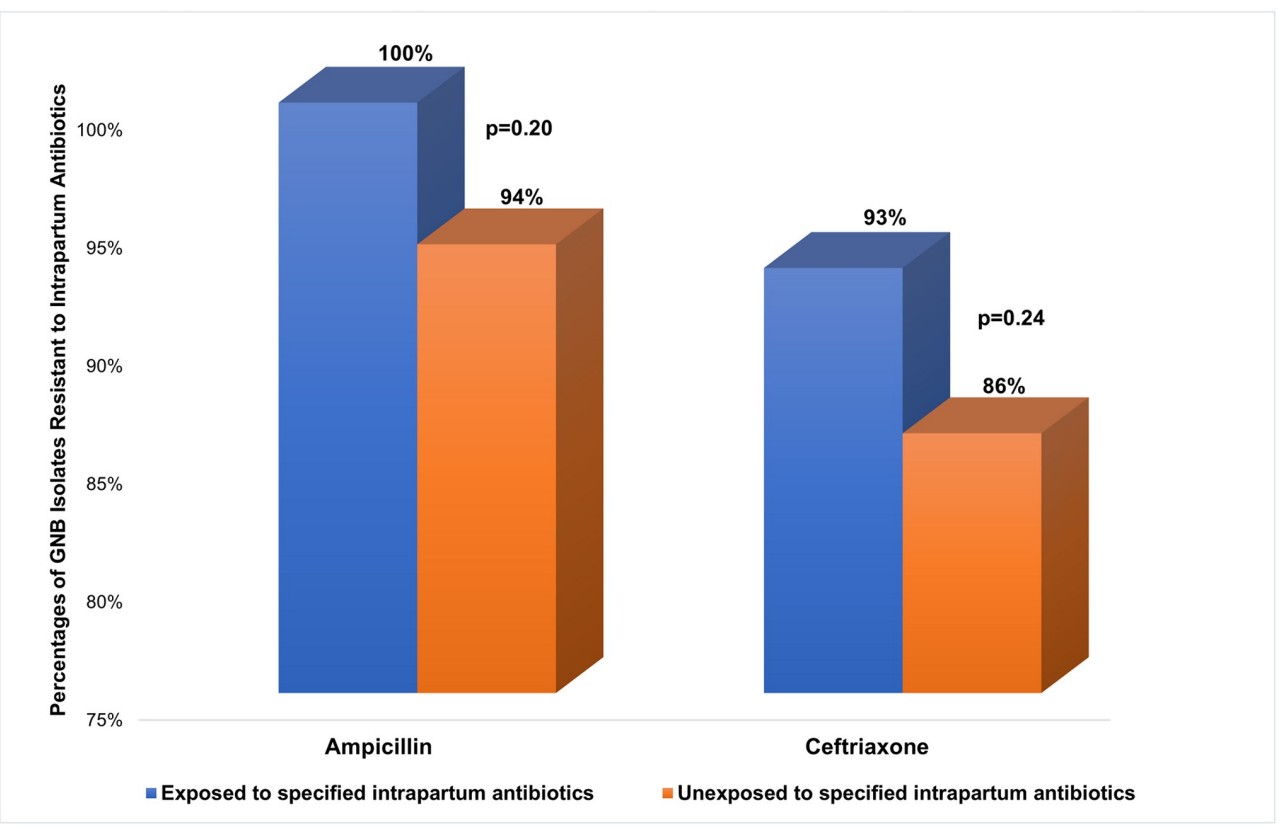

**Fig 13. Intrapartum antibiotics and single-drug resistance.** Ampicillin-resistance was 100% among newborns exposed to intrapartum ampicillin compared to 94% among the unexposed (p = 0.20) while ceftriaxone-resistance was 93% among newborns exposed to intrapartum ceftriaxone compared to 86% among the unexposed (p = 0.24).

of community and hospital-acquired LOS in Ethiopia and identify predominant sources of infections in these settings would help to guide the distribution of resources for interventions towards reducing both LOS and the associated AMR.

The clustered occurrence of some *Klebsiella pneumoniae* (*Kp*) strains suggests possible outbreaks and may provide clues about infection sources in this study. Strains ST35 and ST37, which were responsible for more than half of all GNB sepsis in this study, have been associated with MDR and reported to be sources of outbreaks within NICU settings outside sSA [59–61]. Both ST35 and ST37 have also been detected in the feces of healthy adults and animals in the community [62]. The incidence of ST37 was much higher among newborns that were inborn, delivered via Caesarean section and had LOS, strongly suggesting that majority of ST37 infections were hospital-acquired. Community outbreak and vertical transmission seemed more plausible for ST218 cases, which occurred within a month and mostly among newborns that were outborn, delivered vaginally, and had EOS. With the extreme case fatality rate of 80% observed with ST218 cases, all having the KL57 serotype, it is more likely that the newborns in this study were infected with the hypervirulent type of ST218/KL57, which has been reported among patients at a tertiary hospital about 350km from Addis Ababa in Jimma, Ethiopia, likewise other countries [63, 64]. While there was no difference in the incidence of ST35 between inborn and outborn, the higher incidence of ST35 and ST218 among newborns delivered vaginally may necessitate the screening of pregnant women for colonization by these strains when community outbreaks are suspected. Except for ST37, ST711, and ST985, eight of the 11 *Kp*

**Table 2. Neonatal, maternal, and environmental factors associated with resistance to both ampicillin and gentamicin among 119 newborns with gram-negative sepsis.**

| | Bivariable Analysis | | | Multivariable Model | | | |
|---|---|---|---|---|---|---|---|
| Variable | Risk Ratio | 95% CI | p-value | Variable | Risk Ratio | 95% CI | p-value |
| **Newborns' Characteristics** | | | | | | | |
| Female (ref: male) | 0.91 | 0.78, 1.08 | 0.281 | Female | 0.85 | 0.76, 0.96 | 0.010 |
| Preterm (binary) | | | | | | | |
| <37 weeks (ref: ≥37weeks) | 1.27 | 1.03, 1.56 | 0.024 | | | | |
| Preterm (categorical) | | | | | | | |
| (ref: ≥37weeks) | | | | | | | |
| Moderate/Late (32-<37 weeks) | 1.28 | 1.04, 1.58 | 0.021 | | | | |
| Very/Extremely (<32weeks) | 1.24 | 0.96, 1.60 | 0.102 | | | | |
| Low Birthweight (binary) | | | | | | | |
| <2500g (ref: normal ≥2500g) | 1.35 | 1.03, 1.78 | 0.031 | Low birthweight as binary (<2500g) | 1.36 | 1.02, 1.83 | 0.039 |
| Birthweight (categorical) | | | | | | | |
| (ref: normal ≥2500g) | | | | | | | |
| Low (1500-<2500g) | 1.36 | 1.02, 1.80 | 0.034 | | | | |
| Very Low/Extremely Low (<1500g) | 1.35 | 1.01, 1.80 | 0.040 | | | | |
| Outborn (ref: inborn) | 0.94 | 0.81, 1.10 | 0.451 | | | | |
| Caesarian delivery (ref: vaginal) | 0.97 | 0.82, 1.14 | 0.681 | | | | |
| **Onset of Sepsis** | | | | | | | |
| Late-onset sepsis (ref: early-onset sepsis) | 1.10 | 0.96, 1.27 | 0.175 | Late-onset sepsis | 1.13 | 1.03, 1.23 | 0.007 |
| **Maternal Antibiotics** | | | | | | | |
| Intrapartum Antibiotic (IPA) Exposure (ref: no IPA) | 1.03 | 0.84, 1.26 | 0.811 | | | | |
| **Sanitation and Hygiene** | | | | | | | |
| Type of toilet (ref: flush toilet) | | | | | | | |
| Squat toilet/Pit latrine | 0.93 | 0.70, 1.23 | 0.613 | | | | |
| Communal/Others | 1.04 | 0.77, 1.39 | 0.798 | | | | |
| Access to Running Water (ref: at least 4 days in a week) | | | | | | | |
| Irregular or 2–3 days/week | 1.03 | 0.79, 0.83 | 0.790 | | | | |
| Less than once/week | 1.15 | 0.96, 1.39 | 0.126 | | | | |

strains identified in this study were also found among the Jimma patients, suggesting that some of these strains may be endemic to the region. The BARNARDS study in Ethiopia was limited to a single site; hence, no further comparison could be made with the incidence of *Kp* strains in the community or other NICU settings closer to Addis Ababa. There is a need for surveillance on the K*p* strains responsible for the most neonatal infections and mortality within hospitals and community settings across Ethiopia and sSA. It is imperative to reinforce infection control practices in Ethiopian NICU settings to prevent the spread of deadly *Kp* strains especially because of the limited facilities for early detection.

Intrapartum exposure to ampicillin had been linked with ampicillin-resistant type of neonatal sepsis and this was consistent with findings in our study, not only for ampicillin but also for ceftriaxone, the more frequently used intrapartum antibiotic in this study [35]. There was an increased prevalence of ceftriaxone-resistant GNB isolates with intrapartum exposure to ceftriaxone; however, the higher prevalence of ampicillin and ceftriaxone-resistant GNB sepsis did not reach statistical significance, most likely because of the fewer number of individuals exposed to the antibiotics in this study. Exposure to intrapartum antibiotics was not found to be a significant risk factor for drug resistance to other antibiotics not used as intrapartum antibiotics in this study, consistent with prior studies [37]. Considering that one in four newborns

**Table 3. Neonatal, maternal, and environmental factors associated with multidrug resistance (MDR score of 0–9) among 119 newborns with gram-negative sepsis.**

| | Bivariable Analysis | | | Multivariable Model | | | |
|---|---|---|---|---|---|---|---|
| Variable | β | 95% CI | p-value | Variable | β | 95% CI | p-value |
| **Newborns' Characteristics** | | | | | | | |
| Female (ref: male) | -0.04 | -0.58, 0.51 | 0.892 | | | | |
| Preterm (binary) | | | | | | | |
| <37 weeks (ref: ≥37 weeks) | 0.53 | -0.06, 1.11 | 0.076 | | | | |
| Preterm (categorical) | | | | | | | |
| (ref: ≥37 weeks) | | | | | | | |
| Moderate/Late (32-<37 weeks) | 0.67 | 0.06, 1.28 | 0.032 | | | | |
| Very/Extremely (<32weeks) | 0.05 | -0.80, 0.89 | 0.913 | | | | |
| Low Birthweight (binary) | | | | Low birthweight as binary (<2500g) | 0.81 | 0.17, 1.45 | 0.013 |
| <2500g (ref: normal ≥2500g) | 0.95 | 0.31, 1.59 | 0.004 | | | | |
| Birthweight (categorical) | | | | | | | |
| (ref: normal ≥2500g) | | | | | | | |
| Low (1500-<2500g) | 0.96 | 0.26, 1.66 | 0.007 | | | | |
| Very/Extremely Low (<1500g) | 0.93 | 0.20, 1.66 | 0.013 | | | | |
| Outborn (ref: inborn) | -0.23 | -0.79, 0.33 | 0.419 | | | | |
| Caesarian delivery (ref: vaginal) | -0.16 | -0.63, 0.30 | 0.486 | | | | |
| **Onset of Sepsis** | | | | | | | |
| Late-onset sepsis (ref: early-onset) | 0.60 | 0.02, 1.19 | 0.043 | Late-onset sepsis | 0.59 | 0.02, 1.16 | 0.044 |
| **Maternal Antibiotics** | | | | | | | |
| Intrapartum Antibiotic (IPA) Exposure (ref: no IPA) | 0.10 | -0.66, 0.86 | 0.796 | | | | |
| **Sanitation and Hygiene** | | | | | | | |
| Type of toilet (ref: flush toilet) | | | | | | | |
| Squat toilet/Pit latrine | -0.02 | -1.18, 1.14 | 0.968 | | | | |
| Communal/Others | 0.16 | -1.13, 1.45 | 0.807 | | | | |
| Access to Running Water (ref: at least 4 days/week) | | | | | | | |
| Irregular or 2–3 days/week | 0.25 | -0.46, 0.96 | 0.486 | | | | |
| Less than once/week | 0.21 | -0.55, 0.98 | 0.583 | | | | |

In the multivariable analysis, low birthweight was associated with increased MDR score by 0.81 points while in late-onset sepsis, the MDR score was higher by 0.59 points.

with GNB sepsis was exposed to intrapartum antibiotics, it may be necessary to assess whether the supposed benefit of recommending intrapartum antibiotics outweighs the burden from increased antibiotic-resistant GNB infections among newborns and the rationale for intrapartum antibiotics especially in settings where the there is no evidence for Group B streptococcus early-onset sepsis [39].

A limitation of this study is that we were not able to characterize all GNB isolates during the study due to loss of viability, hence the numbers of GNB total did not match the total WGS. In addition, antimicrobial sensitivity testing was conducted separately for constituents of antibiotic combinations; however, a synergistic effect is unlikely *in vivo* when resistance has been demonstrated to the individual components of a combined therapy *in vitro* [65]. For instance, the high mortality rates observed among newborns treated with AmpGen for AmpGen-resistant GNB sepsis (independent resistance to ampicillin and gentamicin *in vitro*) were similar to those among newborns who received no antibiotics for their GNB infections in this study. Evidence from this study confirmed that newborns are more likely to survive GNB infections if

treated with the appropriate antibiotics. The high case-fatality among newborns with GNB sepsis may be related to ineffective first-line antibiotics.

## Conclusion

The current study revealed a high prevalence of resistance to first-line therapy for neonatal sepsis among GNB isolates in Ethiopia. We found lower resistance to amikacin, piperacillin-tazobactam, and carbapenems. There is a need to reassess the current first-line treatment options with antibiotics that have shown sensitivity against GNB, after they have been evaluated for safety, feasibility, and availability. Stronger microbiology laboratory capacity to diagnose AMR is required in countries with high burden of neonatal sepsis and high case fatality to make clinical and policy decisions. Preterm birth, LBW, and LOS were identified as risk factors for MDR, and these could provide directions for possible interventions directed towards mitigating AMR in neonatal infections. Improving infection control practices, antimicrobial stewardship regarding intrapartum antibiotics and exploring recent alternative therapeutic options to antimicrobials may reduce the prevalence of antibiotic-resistant GNB sepsis among newborns.

## Acknowledgments

We thank the mothers and newborns who participated in BARNARDS-Ethiopia study. We acknowledge the BARNARDS-Ethiopia team Redeat Workneh, Tefere Biteye, Yahya Mohammed, supervisors, and data collectors. We appreciate our colleagues at St. Paul's Hospital Millennium Medical College, Boston Children's Hospital, and Harvard T. H. Chan School of Public Health for their unreserved support during the undertaking of this work, especially Drs. Wendemagen Gezahegn and Balkachew Nigatu. We would like to acknowledge the clinicians and researchers that provided advice and guidance during BARNARDS and the BARNARDS network (https://barnards-group.com). The laboratory work has been coordinated and completed with Cardiff University and we would like to acknowledge Ana Ferreira, Edward Portal, Calie Dyer, and Jordan Mathias. We thank Madeline Van Husen for her assistance in formatting the tables, figures, and manuscript.

## Author Contributions

**Conceptualization:** Semaria Solomon, Delayehu Bekele, Grace J. Chan.

**Data curation:** Oluwasefunmi Akeju, Oludare A. Odumade, Kimi Van Wickle, Maria J. Carvalho, Kirsty Sands.

**Formal analysis:** Oluwasefunmi Akeju, Kimi Van Wickle, Maria J. Carvalho, Kathryn Thomson, Timothy R. Walsh.

**Investigation:** Semaria Solomon.

**Methodology:** Oluwasefunmi Akeju, Kimi Van Wickle, Frederick G. B. Goddard, Grace J. Chan.

**Project administration:** Semaria Solomon, Rozina Ambachew, Zenebe Gebreyohannes, Mahlet Abayneh, Gesit Metaferia, Rebecca Milton.

**Resources:** Semaria Solomon, Oludare A. Odumade, Rozina Ambachew, Zenebe Gebreyohannes, Mahlet Abayneh, Gesit Metaferia.

**Supervision:** Semaria Solomon, Delayehu Bekele, Grace J. Chan.

**Validation:** Kimi Van Wickle, Rebecca Milton.

**Visualization:** Frederick G. B. Goddard.

**Writing – original draft:** Oluwasefunmi Akeju, Oludare A. Odumade, Frederick G. B. Goddard.

**Writing – review & editing:** Semaria Solomon, Grace J. Chan.

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
