## [Decision Letter · Decision Letter 0]

16 Apr 2021

PONE-D-21-07720

Prevalence and risk factors for antimicrobial resistance among newborns with gram-negative sepsis

PLOS ONE

Dear Dr. Chan,

Thank you for submitting your manuscript to PLOS ONE. After careful consideration, we feel that it has merit but does not fully meet PLOS ONE’s publication criteria as it currently stands. Therefore, we invite you to submit a revised version of the manuscript that addresses the points raised during the review process.

Please address the concerns that the reviewers highlighted and submit the manuscript as early as your convenience

We look forward to receiving your revised manuscript.

Kind regards,

Monica Cartelle Gestal, PhD

Academic Editor

PLOS ONE

Journal Requirements:

3. We note that you are reporting an analysis of a microarray, next-generation sequencing, or deep sequencing data set. PLOS requires that authors comply with field-specific standards for preparation, recording, and deposition of data in repositories appropriate to their field. Please upload these data to a stable, public repository (such as ArrayExpress, Gene Expression Omnibus (GEO), DNA Data Bank of Japan (DDBJ), NCBI GenBank, NCBI Sequence Read Archive, or EMBL Nucleotide Sequence Database (ENA)). In your revised cover letter, please provide the relevant accession numbers that may be used to access these data. For a full list of recommended repositories, see http://journals.plos.org/plosone/s/data-availability#loc-omics or http://journals.plos.org/plosone/s/data-availability#loc-sequencing.

Additional Editor Comments (if provided):

Reviewers' comments:

Reviewer's Responses to Questions

**Comments to the Author**

1. Is the manuscript technically sound, and do the data support the conclusions?

Reviewer #1: Yes

Reviewer #2: Yes

2. Has the statistical analysis been performed appropriately and rigorously? 

Reviewer #1: Yes

Reviewer #2: Yes

3. Have the authors made all data underlying the findings in their manuscript fully available?

Reviewer #1: Yes

Reviewer #2: Yes

4. Is the manuscript presented in an intelligible fashion and written in standard English?

Reviewer #1: Yes

Reviewer #2: Yes

5. Review Comments to the Author

Reviewer #1: A well designed study.

How frequently does the study hospital change its antibiotic policy?

The effective sample size was 119. Total number of newborn enrolled i.e. 4828 was not used for analysis. Therefore, it may not be included in abstract.

Reviewer #2: Please see the attached document with my comments, suggestions, and questions. Please consider these and answer the questions as best you can.

A note about the figures and tables:

The authors are encouraged to make a list all figures at the end of the document. Moreover, include figure captions, or legends, that are complete so that the reader is able to understand the figure without reference to the text. Describe only what is seen on the image. Use of arrows, asterisks, arrowheads, etc. are useful to communicate or covey to the readers your findings. The authors are advised to revise the captions all figures.

The authors are encouraged to make a list all tables at the end of the document. The headings or titles of the tables should be complete so that the reader is able to understand the table without reference to the text. The authors are advised to revise the titles of all tables. For example: “Table 1.Prevalence and risk factors for antimicrobial resistance among newborns with Gram-negative sepsis: Descriptive analysis for baseline characteristics of 119 newborns Included in Antimicrobial Resistance Study”

6. PLOS authors have the option to publish the peer review history of their article (what does this mean?). If published, this will include your full peer review and any attached files.

Reviewer #1: **Yes: **Manas Pratim Roy

Reviewer #2: **Yes: **Uriel Blas-Machado

---

## [Author Response · Author response to Decision Letter 0]

7 Jun 2021

Thank you for taking the time to comment on this manuscript. Your comments are highly constructive and appreciated.

Reviewer #1

1. How frequently does the study hospital change its antibiotic policy?

Response: Thank you, changes to antibiotic policy in the study hospital are made based on recommendations and policy changes from Ministry of Health and does not occur regularly, the last change to antibiotic policy occurred in 2014.

2. The effective sample size was 119. Total number of newborn enrolled i.e. 4828 was not used for analysis. Therefore, it may not be included in abstract.

Response: Thank you, this was removed in the abstract.

Reviewer #2

1. Please see the attached document with my comments, suggestions, and questions. Please consider these and answer the questions as best you can

Response: Thank you for comments and suggestions. We have addressed them as best we can. Following recommendations from CDC website and Cochrane, "Gram should be capitalized and never hyphenated when used as Gram stain; gram negative and gram positive should be lowercase and only hyphenated when used as a unit modifier", we did not capitalize the “g” in gram-negative but accepted all other suggested edits. 

2. A note about the figures and tables:

The authors are encouraged to make a list all figures at the end of the document. Moreover, include figure captions, or legends, that are complete so that the reader is able to understand the figure without reference to the text. Describe only what is seen on the image. Use of arrows, asterisks, arrowheads, etc. are useful to communicate or covey to the readers your findings. The authors are advised to revise the captions all figures.

Response: The figure legends and captions have now been revised as advised. The figures are cited in the text in order per the PLOSone formatting instructions and uploaded as sperate individual figure files. https://journals.plos.org/plosone/s/file?id=wjVg/PLOSOne_formatting_sample_main_body.pdf, we have added a list of all figures to the end of the document as suggested. We understand colors are typically part of the legend, however we kept the legends defining the colors used in the figure in the figures in case this is easier to format in the publication. Please let us know if you’d prefer that we move these into the manuscript. 

The authors are encouraged to make a list all tables at the end of the document. The headings or titles of the tables should be complete so that the reader is able to understand the table without reference to the text. The authors are advised to revise the titles of all tables. For example: “Table 1.Prevalence and risk factors for antimicrobial resistance among newborns with Gram-negative sepsis: Descriptive analysis for baseline characteristics of 119 newborns Included in Antimicrobial Resistance Study”

Response: The table titles and legends have now been revised as advised. We have added a list of tables to the end of the document as per the reviewer’s comment. 

3. Why 6 drops and not 5 or 7? What is the reasoning? Please explain.

During protocol development 6 drops was agreed as a guide to allow sufficient volume of blood to adhere to the agar and encourage microbial growth. It might be best to use ‘aliquot’ rather than being too detailed, hence the wording has been changed accordingly.

Thank you your time and helpful comments to strengthen the manuscript.

---

## [Decision Letter · Decision Letter 1]

16 Jul 2021

Prevalence and risk factors for antimicrobial resistance among newborns with gram-negative sepsis

PONE-D-21-07720R1

Dear Dr. Chan,

We’re pleased to inform you that your manuscript has been judged scientifically suitable for publication and will be formally accepted for publication once it meets all outstanding technical requirements.

Kind regards,

Monica Cartelle Gestal, PhD

Academic Editor

PLOS ONE

Additional Editor Comments (optional):

Reviewers' comments:

Reviewer's Responses to Questions

**Comments to the Author**

1. If the authors have adequately addressed your comments raised in a previous round of review and you feel that this manuscript is now acceptable for publication, you may indicate that here to bypass the “Comments to the Author” section, enter your conflict of interest statement in the “Confidential to Editor” section, and submit your "Accept" recommendation.

Reviewer #1: All comments have been addressed

Reviewer #2: All comments have been addressed

2. Is the manuscript technically sound, and do the data support the conclusions?

Reviewer #1: Yes

Reviewer #2: (No Response)

3. Has the statistical analysis been performed appropriately and rigorously? 

Reviewer #1: Yes

Reviewer #2: (No Response)

4. Have the authors made all data underlying the findings in their manuscript fully available?

Reviewer #1: Yes

Reviewer #2: (No Response)

5. Is the manuscript presented in an intelligible fashion and written in standard English?

Reviewer #1: Yes

Reviewer #2: (No Response)

6. Review Comments to the Author

Reviewer #1: Changes are satisfactory. However, if the hospital has not changed its policy on antibiotic use after 2014, what would be the practical implication of such a study?

Reviewer #2: (No Response)

7. PLOS authors have the option to publish the peer review history of their article (what does this mean?). If published, this will include your full peer review and any attached files.

Reviewer #1: **Yes: **Manas Pratim Roy

Reviewer #2: **Yes: **Uriel Blas-Machado

---

## [Editor Report · Acceptance letter]

23 Jul 2021

PONE-D-21-07720R1 

Prevalence and risk factors for antimicrobial resistance among newborns with gram-negative sepsis 

Dear Dr. Chan:

I'm pleased to inform you that your manuscript has been deemed suitable for publication in PLOS ONE. Congratulations! Your manuscript is now with our production department. 

Kind regards, 

on behalf of

Dr. Monica Cartelle Gestal 

Academic Editor

PLOS ONE